# Predicting Wrist Joint Angles from the Kinematics of the Arm: Application to the Control of Upper Limb Prostheses

**DOI:** 10.3390/biomimetics8020219

**Published:** 2023-05-24

**Authors:** Antonio Pérez-González, Victor Roda-Casanova, Javier Sabater-Gazulla

**Affiliations:** Department of Mechanical Engineering and Construction, Universitat Jaume I, 12071 Castellón de la Plana, Spain; vroda@uji.es (V.R.-C.); jsabater@uji.es (J.S.-G.)

**Keywords:** wrist joint, wrist motion, arm joints, arm motion, kinematics, upper limb prosthesis, artificial neural network, pick and place, prosthesis control

## Abstract

Automation of wrist rotations in upper limb prostheses allows simplification of the human–machine interface, reducing the user’s mental load and avoiding compensatory movements. This study explored the possibility of predicting wrist rotations in pick-and-place tasks based on kinematic information from the other arm joints. To do this, the position and orientation of the hand, forearm, arm, and back were recorded from five subjects during transport of a cylindrical and a spherical object between four different locations on a vertical shelf. The rotation angles in the arm joints were obtained from the records and used to train feed-forward neural networks (FFNNs) and time-delay neural networks (TDNNs) in order to predict wrist rotations (flexion/extension, abduction/adduction, and pronation/supination) based on the angles at the elbow and shoulder. Correlation coefficients between actual and predicted angles of 0.88 for the FFNN and 0.94 for the TDNN were obtained. These correlations improved when object information was added to the network or when it was trained separately for each object (0.94 for the FFNN, 0.96 for the TDNN). Similarly, it improved when the network was trained specifically for each subject. These results suggest that it would be feasible to reduce compensatory movements in prosthetic hands for specific tasks by using motorized wrists and automating their rotation based on kinematic information obtained with sensors appropriately positioned in the prosthesis and the subject’s body.

## 1. Introduction

Manipulation is a constant and necessary activity in daily life, so the loss of a hand or arm is a serious disability. Hand prostheses can overcome some of the limitations. However, despite technological advances in myoelectric prostheses in recent years, many users abandon their use due to discomfort or dysfunctional control [1,2]. Currently, most upper limb prostheses function with a fixed wrist joint or with a single passive degree of freedom: pronation/supination (P/S) or flexion/extension (F/E). Active control of one or more degrees of freedom in the wrist complicates mechanical design, makes control difficult, and increases the user’s cognitive load. However, the absence of the wrist joint requires compensatory movements of the arm or body during manipulation, movements that are uncomfortable, make some tasks difficult, and can generate musculoskeletal problems in the long term.

The importance of wrist mobility in upper limb prostheses has been demonstrated in some previous work in the literature [3], in which the relative importance for manipulation of degrees of freedom in the wrist compared to those in the hand itself was analyzed. It was concluded that it is possible to simplify the design of the prosthetic hand if it is compensated for with greater mobility in the wrist. However, simultaneous control of the wrist and hand in a prosthetic device is complex. In the vast majority of current commercial prostheses, this control is performed from surface electromyography (EMG) signals in the forearm. The control method is usually sequential, with a pair of electrodes or channels that control, respectively, the flexion and extension movements of the wrist or the opening and closing movements of the hand, with simultaneous activation of both channels used as a mode to switch control between hand and wrist [4]. This control method is cumbersome and slow when movements involve the activation of several degrees of freedom, so, in recent decades, research has been carried out on other methods, such as pattern recognition through the simultaneous use of multiple EMG sensors on the forearm or regression between EMG signals and arm kinematics [4,5], in order to approach simultaneous and proportional control of several degrees of freedom. However, a common problem with EMG-based control methods is that they suffer from reliability and repeatability issues due to multiple factors, such as inadequate sensor contact with the skin, the change in the position of the contact point during manipulation, or the effect of sweat on the EMG signal [1]. Alternatively, other options for the type of source signal used in the simultaneous control of several degrees of freedom, such as ultrasound [6,7] or electrical impedance tomography [8], have been studied at the laboratory level. However, these options are not yet available at the clinical level.

All the control methods mentioned above use physiological signals generated from the activation of the arm or forearm muscles to predict the movement intention of the hand or wrist. In this work, it is hypothesized that, in some tasks, it is possible to relate the movement of the wrist to the movement performed in other joints of the arm, which would allow predicting the movement of the wrist from other kinematic parameters of the arm. For example, in tasks where an object must be picked up and moved to a different position on a shelf, wrist movements are related to the position of the object in the medial-lateral or cranial-caudal directions. For instance, to pick up a cylindrical object resting on a horizontal surface located below the elbow, the wrist needs to be slightly adducted, whereas to position the same object on another horizontal surface located at head height, the wrist needs to be abducted. Flexion and extension movements of the wrist also differ depending on whether the object is to the left or right of the sagital plane passing through the shoulder of the dominant arm. Another task in which wrist movement can be easily related to the rest of the arm is driving a vehicle because, depending on the position of the arm, it is possible to predict the necessary wrist movement to turn the steering wheel or shift gears. In short, it seems feasible to automate wrist movement for certain tasks based on the movement of other arm joints. The acquisition of kinematic signals using inertial sensors (IMUs) has undergone significant advances in recent decades [9], so wrist control based on this type of signal could be robust, simple, and an alternative or complementary to the use of EMG or other more complex techniques, such as ultrasound or tomography.

From a mathematical perspective, the estimation of wrist movements based on different types of recorded signals (EMG, kinematics, etc.) has been approached with different perspectives in the recent literature. Some works focus on signal classification techniques to assign them to a discrete movement or a specific wrist angle. For example, Fajardo-Perdomo et al. [10] used techniques such as support vector machine (SVM) and multilayer perceptron (MLP) neural networks to classify five different static positions between the maximum flexion and extension of the wrist based on EMG signals recorded in the forearm. Yang et al. [7] used dimensional reduction techniques, such as subclass discriminant analysis (SDA) and principal component analysis (PCA), to classify different hand postures and wrist pronation/supination based on ultrasound signals. Dynamic estimation of wrist angles represents a qualitative advance over the previous approach. It was approached by Xie et al. [6] based on ultrasound signals. They used SVM and backpropagation neural network (BPNN) techniques to predict the wrist extension angle adequately in various extension cycles from the neutral position at different frequencies. Liu et al. [11] used a recurrent neural network with hidden states, including long short-term memory (LSTM) cells, to predict wrist position in space during assembly tasks, anticipating up to 2 s based on the previous movement obtained using depth cameras. Zheng et al. [8] used linear regression methods, such as the least absolute shrinkage and selection operator (LASSO) and support vector regression (SVR), to estimate the dynamic movement of the wrist based on electrical impedance tomography signals. Qin et al. [5] used convolutional neural networks (CNNs) similar to those used in image processing to process the amplitude information from a matrix of 32 EMG sensors as images and predict dynamic movements that included a sequence of flexion/extension and pronation/supination of the wrist and opening and closing of the hand. They compared predictions with those obtained by other regression methods, such as linear regression, k-nearest neighbors (KNNs), SVR, and decision trees, confirming a better correlation with the proposed CNN-based method. Casini et al. [12], on the other hand, used PCA to obtain the principal components of the human wrist rotations during gripping actions with a variety of objects to implement the determined synergies in an underactuated prosthetic wrist. In this case, the focus was not on estimating wrist rotations but on the dimensional reduction of its common movements.

In this work, we analyzed the possibility of predicting wrist movement in certain tasks based on arm kinematics using neural networks and linear regression models. To this end, we used experimental data for position and orientation recorded from five subjects performing an object transport task (pick and place) in which two different objects were relocated between several positions on a vertical shelf. The kinematic data were processed to obtain angles between arm joints, and then neural networks were trained to anticipate the required wrist rotation angles based on other kinematic data for the arm recorded prior to the required instant. Different network architectures and input parameters were analyzed, as well as the effect of temporal anticipation on the predictive capacity of the network.

## 2. Materials and Methods

### 2.1. Experiments

Five healthy volunteers without any known upper limb pathology participated in an object manipulation experiment following a protocol approved by the Ethics Committee of Universitat Jaume I (reference CD/37/2022). The subjects were university students, with a mean age of 20.4 years (standard deviation 1.5) and mean height of 174.8 cm (standard deviation 6.1). All of them were right-handed males except for one who was ambidextrous. The subjects were equipped with different sensors using the Polhemus Fastrak motion capture system, which allows the position and orientation of up to four sensors to be instantaneously recorded with respect to a fixed transmitter. Four sensors were used placed in the following positions on the subject (Figure 1): hand, forearm, arm, and trunk (back). The arm sensors were sewn onto elastic Tubigrip® bands placed on the subject’s right arm, forearm, and hand. The trunk sensor was attached with Velcro to a posture-correcting harness for the back. The arm sensors were positioned so that the *x*-axis of the sensor pointed in the distal direction and that of the back was directed towards the ground. The Fastrak system transmitter was fixed on the horizontal upper surface of the shelf in the back left corner so that the reference *x*-axis pointed towards the subject’s right, the *y*-axis towards the subject, and the *z*-axis downwards (see Figure 2). With the chosen setup, all sensors moved during the experiment in the x>0 hemisphere of the Fastrak system transmitter.

The subjects performed transport tasks with the two lightweight abstract objects from the Southampton Hand Assessment Protocol (SHAP) [13] corresponding to power cylindrical grip (cylinder with a diameter of 50mm, height of 100mm, and mass of 18g) and spherical grip (sphere with a diameter of 70mm and mass of 26g) (objects shown in Figure 3). The object transport tasks were performed from a starting position to a target position, with the starting and target positions being the different combinations of the four possible positions (UL, UR, DL, DR) of a vertical shelf placed on a table in front of the subject, as indicated in Figure 2.

At the beginning of the experiment, the subject was positioned in the anatomical position (Figure 1). The object to be transported (cylinder or sphere) was located in one of the four possible positions (UL, UR, DL, DR). At the experimenter’s command, the subject made a natural arm movement from the anatomical position, without moving their feet, to pick up the object from its starting position, transport it to the predetermined target position indicated by the experimenter prior to the trial, release it in that position, and return the arm to the anatomical starting position. This task was performed three times consecutively with the same starting and target positions for the object after some trial repetitions prior to recording the experiment. The experiment was then repeated but with the starting and target positions of the object changed. The transport tasks were performed for all 12 possible combinations between the four positions (UL, UR, DL, DR) taken two at a time (origin, destination), with three repetitions of each combination. The cylindrical object was placed with its axis vertical in the starting and destination positions. The trials with each subject were all conducted in the same session and the order of the transports was the same for all subjects, with movements with the cylinder being performed first and then with the sphere. A short break of a few minutes was allowed between trials with each object.

### 2.2. Treatment of Experimental Records

The data recorded by the Fastrak system sensors were processed using Matlab. Initially, the data from each sensor in each repetition were obtained. These data consisted of the *x*, *y*, and *z* coordinates of each sensor relative to the transmitter and the three Euler angles of orientation of the sensor relative to the transmitter. The Euler angles were subsequently properly processed to avoid abrupt jumps since the values were recorded in the range between $-180^{\circ }$ and $+180^{\circ }$. Subsequently, synchronization of the measurements recorded in each of the three repetitions of the experiment was performed in order to obtain the same duration for each repetition. For this purpose, the dynamic time warping algorithm *dtw* from the “Signal Processing Toolbox” of Matlab was used. Then, all the measurements were filtered with a third-order Butterworth low-pass filter using the *butter* function of Matlab from the same toolbox. The final filtered and synchronized data were interpolated and time was normalised to represent each trial with 101 points evenly distributed between the start and end of the trial corresponding to normalised time steps of 0.01. Since the average duration of a trial was around 5 s, each relative time increment corresponded to approximately 50 ms. For each instant of time, the three Euler angles corresponding to the rotation between the sensors located on both sides of each joint were calculated. These angles were approximations of the actual anatomical angles in these joints. Specifically, the rotation angles were calculated in the wrist for rotation between sensor one (hand) and sensor two (forearm), in the elbow and the pronation/supination of the forearm for rotation between sensor two (forearm) and sensor three (arm), and in the shoulder from the rotations between sensor three (arm) and sensor four (trunk). For each joint, the order of the rotations performed for the calculation of the angles was the rotation corresponding to flexion/extension (F/E) followed by abduction/adduction (A/A) and, finally, pronation/supination (P/S).

### 2.3. Neural Networks

The F/E and A/A angles in the wrist and the P/S in the forearm were taken as the objective angles to be predicted by the neural networks in this study since the main P/S movement for the hand is performed in the forearm and the P/S rotation in the wrist joint itself is negligible. The remaining two angles at the elbow joint (F/E and A/A) and the three at the shoulder were used as input data.

The training and testing of the neural networks was performed using Matlab. Two different approaches were considered with different levels of complexity, using a simple feed-forward neural network (FFNN) in one case and a time-delay neural network (TDNN) in the other. For this purpose, the *feedforwardnet* and *timedelaynet* functions of Matlab’s “Deep Learning Toolbox” were used, respectively.

Since two different objects were moved in the trials performed involving different grip modes and also predictably different wrist movements, the effects of training the neural networks with the experiments on each object separately or considering all of them together were analyzed. Similarly, the effects of training the network for a specific subject or jointly including all subjects were studied. Moreover, the effect of the number of neurons in the hidden layer on the predictive capacity was also analyzed, as well as the effect of the training algorithm among the several available in Matlab. Specifically, for the number of neurons in the hidden layer, values between 5 and 20 were analyzed, considering that the number of cases available for training was 720 (5 subjects × 12 trials × 2 objects × 3 repetitions × 2 instants) and that there are recommendations for an optimal number of neurons around the logarithm of base 2 of the number of cases [14]. A random percentage of 15% of cases was selected for the final test of the network, while the rest were used for training. For other parameters of the networks and their training, the default values of the Matlab functions were taken. To analyze the possible effect of randomness in the selection of the test cases, the results shown were obtained as the average of 5 different training sessions for the network.

#### 2.3.1. Feed-Forward Neural Network (FFNN)

In the case of the FFNN, in order to limit the problem, the intended output data were the angles of the wrist only at the instants of picking up the object (pick, PK) and releasing it (place, PL), and the input data were taken at a specific instant of the movement anticipated with respect to the instants PK and PL. The justification was that, in this way, in a subsequent application to a wrist prosthesis, this anticipation would allow the motorized wrist to rotate during the time lag between the instant of measurement of the input data and the desired final posture. The instants PK and PL were identified from the minimum values of the Y coordinate of the hand sensor position in the parts of the record prior to the transport movement (for PK) and subsequent to it (for PL). In order to minimize the effects of differences due to sensor placement on different subjects, the angles for each subject were recalculated and introduced to the network as differences with respect to the mean value of that angle in all the experiments with the same subject.

The effect on the predictive ability of the neural network of the anticipation between the input data and the PK and PL moments was studied. For this purpose, the prediction of angles for PK was estimated by taking the input data with a time anticipation of 0.5s or 0.75s with respect to the PK instant and, in the same way, for the PL instant. Furthermore, the effect of introducing the task (PK or PL) as an input to the FFNN was analyzed.

A shallow network with a single hidden layer was used as architecture for the FFNN in order to keep it simple. Figure 4 shows the schematic representation of the network for a case with 20 neurons in the hidden layer, 5 input scalar parameters corresponding to the F/E and A/A angles in the elbow and the thee angles in the shoulder, and 3 output scalar parameters corresponding to the F/E and A/A angles in the wrist and the P/S angle in the forearm.

#### 2.3.2. Time-Delay Neural Network (TDNN)

Time-delay networks allow dynamic estimations based on a set of data collected at several instants prior to the prediction and are common in time-series prediction. Figure 5 shows the schematic representation of the neural network for a case with 10 neurons in the hidden layer. The five input time series corresponded to the F/E and A/A angles at the elbow and the three shoulder angles and the output time series corresponded to one of the angles at the wrist. Figure 5 shows that the input included data corresponding to the previous 10 instants of the time series (1:10) amounting to an interval of approximately 0.5s amplitude. The effect of varying the delay was studied through different alternative options: 15 instants earlier (1:15) than the target instant, equivalent to about 0.75s; 2 instants earlier (1:2), equivalent to 0.1s; and 2 instants anticipated 0.5s earlier than the target instant (11:12). Fifteen percent of the cases were randomly selected for testing, half corresponding to the cylinder transport and half to the sphere, and the remaining cases were used for training.

Due to limitations of the Matlab function used, the time series of the inputs and outputs of the network corresponding to the different experiments of each group (training and testing) were concatenated as a single time series. In order to eliminate abrupt jumps in this concatenation, a Tukey-type window was applied to the data for each included series using the *tukeywin* function of Matlab’s “Signal Processing Toolbox”. This function multiplies the first and last data points of a series by a cosine function to bring it to zero at the start and end points, preserving the central data of the series unaltered. The modified data included the first five and last five instants from the series of 101 data points from each experiment corresponding to the moments of transition of the arm from and to the anatomical resting position, thus having a minimal effect on the results of interest.

### 2.4. Linear Regression Models

As an alternative to the neural networks, the use of linear regression models to predict the wrist angles from the other joint angles was tested. For this purpose, the *fitlm* function of Matlab’s “Statistics and Machine Learning Toolbox” was used. A linear model was fitted to each of the target angles (F/E and A/A angles in the wrist and the P/S in the forearm) from the remaining two angles at the elbow joint (F/E and A/A) and the three at the shoulder. The fitting was based on the whole set of data. For this, a single matrix of angles was vertically stacked with the information from all the experiments so that each row corresponded to the target and predictor angles for an instant in a particular experiment. The correlation coefficient between the target angles and the model prediction and the mean absolute errors were taken as indicators for the goodness of fit.

## 3. Results

Figure 6 shows an example of the angles at the wrist for subject one in the movement from UL to DL for both objects (cylinder and sphere). The three repetitions and the mean value are shown, indicating the good repeatability of the measurements. The results showed different movements at the wrist for each object, with more pronounced differences in the F/E pattern and in the amplitude of the P/S movements. The movement with the largest range was the P/S movement and the smallest range was the A/A movement. The mean range of amplitude for these angles averaged across subjects during all the experiments was $29.2^{\circ }$ for F/E, $20.7^{\circ }$ for A/A, and $92.9^{\circ }$ for P/S.

Figure 7 shows the angles for F/E and A/A at the elbow and the three shoulder angles for the same movements and the same subject as in Figure 6. The good repeatability of the results and different patterns depending on the manipulated object can also be seen. For other starting and target points and for other subjects, different movement patterns were obtained but with similar repeatability.

### 3.1. Feed-Forward Neural Network (FFNN)

Table 1 shows the mean errors as absolute values for each of the wrist angles obtained from the difference between the actual data recorded at the wrist and those predicted by the FFNN with the training function *trainbr*, using as input data for the network the angles for F/E and A/A at the elbow and the three shoulder angles with 0.5s and 0.75s of anticipation with respect to the PK and PL instants and different numbers of neurons in the hidden layer. The correlation coefficient (CC) between the actual angles and the prediction is also shown. The results correspond to the averages from five repetitions of the training and testing of the network with the whole set of data corresponding to the five subjects. The average standard deviation between the different repetitions was close to 7.5% for each data point, so the result was considered to be sufficiently representative. The highest error in prediction was observed in the P/S angle, with lower values in the F/E and A/A angles, indicating a certain correlation between the error in prediction and the range of variation in the corresponding angle in the trials.

Figure 8 shows the correlation coefficient between the actual angle values and those estimated by the network as a function of the number of neurons in the hidden layer and the anticipation in time between the data and the prediction. The effect of including the information about the manipulated object or subject in the input data for the network for an anticipation of 0.5s is also shown. An improvement in correlation was observed with an increase in the number of neurons, stabilizing for values between 15 and 20 neurons. The differences between data anticipation of 0.5s and 0.75s with respect to the prediction were small. Including subject information as input for the network improved the prediction, even more so with the inclusion of the object, resulting in correlation coefficients close to 0.95.

When the task information (PK or PL) was added to the network as an input, the changes in the CC were negligible. For example, for anticipation of 0.5s and 20 neurons, the CC changed from 0.883 to 0.887 and the corresponding changes in the mean error for the angles were also small.

Figure 9 shows an example of the comparison between the actual angles and the wrist angles predicted by the network for anticipation of 0.5s and 20 neurons with and without inclusion of object information as input for the network.

Figure 10 shows examples of the variation in the loss function (mean squared error) during the training process for the FFNN with anticipation of 0.5s and 20 or 10 neurons in the hidden layer. The loss stabilized to a nearly constant value after less than 100 epochs. A greater difference between test and train values for 20 neurons than for 10 neurons can be observed, which may indicate some degree of overfitting for 20 neurons.

The training function used to train the FFNN had an effect on its predictive capacity. Among the various functions available in Matlab, the function that gave the best results was *trainbr*. It is a function with Bayesian regularization backpropagation that updates weight and bias values through Levenberg–Marquardt optimization and selects an optimal combination in the minimization of weights and mean squared errors to achieve a well-generalized network. Table 2 shows the results obtained for the correlation coefficient and mean prediction errors with different training functions in the case of 0.5s anticipation, 20 neurons in the hidden layer, and no inclusion of object information.

### 3.2. Time-Delay Neural Network (TDNN)

Figure 11 shows a randomly selected example of the results for the transport of each object predicted with the TDNN with the training function *trainbr* using as input data the angles for F/E and A/A at the elbow and the three angles of the shoulder and comparing the actual value (target) with the prediction of the network (output). In this example, the networks used, which differed for each output angle, included 15 neurons and predicted the wrist angle for each instant from the input data recorded in the previous 15 instants (equivalent to a time delay of approximately 0.75s). The results show that the predictions exhibited, in general, a larger oscillation than the actual curve, but they approximated the evolution of the actual curves well.

Table 3 compares the results in terms of the error in the prediction of each angle by the TDNN (root-mean-square error) and the mean correlation coefficient for different numbers of neurons and amplitudes of temporal anticipation with the training function *trainbr*. As for the FFNN, the amplitude of the error was greater for the P/S angle, which was also the one with the greatest range of variation during the trials.

Figure 12 shows the change in the correlation coefficient between the actual and network-predicted results for all time series from the test group cases as a function of the number of neurons in the hidden layer and the time delay amplitude used (0.1s, 0.5s, or 0.75s). There was a tendency for the correlation to stabilize around 15–20 neurons. Similarly, increasing the amplitude of the time delay also improved the prediction, although the improvement between 0.5s and 0.75s was very limited. If the input data had an amplitude of 0.1s but were 0.5s ahead of the prediction instant, the prediction deteriorated, lowering the correlation coefficient by about 0.1 (not shown in the figure). The figure also shows, for the time delay amplitude of 0.5s, the effect of restricting the dataset to data corresponding to each object separately (the average of both objects is shown with differences between them being less than 0.02 in terms of the correlation coefficient) or to data corresponding to a single subject (the result for subject one is shown). In both cases, the restriction of the dataset used allowed the improvement of the correlation coefficient.

Figure 13 shows the variation in the loss function (mean squared error) during the training process for the TDNN with anticipation of 0.5s and 20 or 10 neurons in the hidden layer. The loss stabilized to a nearly constant value after fewer than 100 epochs. The difference between 20 or 10 neurons was small, but some degree of overfitting for 20 neurons could be inferred from the lower loss value for the train set than for the test set. A very good fit was observed for 10 neurons.

Table 4 shows the results obtained for the error in the angles and the mean correlation coefficient for the different training functions available for the TDNN in the case of a 0.5s delay amplitude and 20 neurons in the hidden layer. The best result was obtained with the *trainbr* function, as was the case for the FFNN.

### 3.3. Linear Regression Models

Table 5 shows the performance of the linear regression models fitted considering the data from all the experiments. The mean absolute errors in the angles were greater and the CCs lower than those obtained with the FFNN and TDNN. A higher CC was obtained for the P/S angle compared to the F/E and A/A angles.

## 4. Discussion

This work analyzed the feasibility of using neural networks for predicting necessary wrist movements in object transport tasks based on kinematic information from the rest of the arm. Wrist movement is important in upper limb prosthetic devices to reduce compensatory arm movements. Estimating or predicting necessary wrist movement from other parameters would facilitate automatic or semi-automatic operation of upper limb prostheses, reducing users’ mental fatigue. Other researchers have attempted to predict wrist movement from signals associated with muscle activity in the forearm, such as EMG signals or ultrasound images of the forearm area, using techniques such as neural networks and other classification and regression techniques, including SDA, PCA, KNN, and SVM. However, EMG and ultrasound-based systems are complex, difficult to train, and subject to sensor movement failure.

The results of this study indicate that it is feasible to train neural networks to predict, with acceptable errors, required rotation angles in the human wrist during object transport tasks based on rotations in more proximal joints (shoulder, elbow). The prediction errors achieved were around 10–15% of the wrist movement range for both the FFNN and TDNN. The CCs between actual and estimated values reached levels greater than 0.85 for the FFNN and greater than 0.94 for the TDNN. Predictive capacity improved when training was customized for the subject or when the network also had information about the manipulated object. These values were comparable to or better than results obtained in the literature using EMG signals. In ref. [10], classifications of wrist flexion positions were achieved with 77% accuracy using neural networks and 53% accuracy using SVM. In ref. [5], correlation coefficients between 0.87 and 0.95 were achieved, depending on the subject, when predicting simple F/E and P/S wrist movements and hand-opening/closing using deep CNNs more complex than those used in this work. In studies using ultrasound signals, such as in ref. [8], $R^{2}$ determination coefficients of 0.92 were obtained for simple wrist movements using LASSO and SVR regression techniques. In this study, we also tested the use of linear regression models to relate the wrist angles to the angles of the other arm joints, but the results indicated low CCs for those models. These results suggest that neural networks are much better than linear regression models for this purpose.

In this study, two different strategies were tested to address the problem of wrist posture prediction from arm kinematics through neural networks with different levels of complexity and network sizes.

The first strategy consisted of the use of a feed-forward neural network (FFNN), which was used to predict the wrist posture only at the instants of object picking (PK) or object placement (PL) from the elbow and shoulder angles recorded some time in advance of those instants. The practical implementation of this strategy would require recording the input data for the network for the anticipated instant and using them to rotate the wrist at the mentioned instants (PK, PL). These instants could be detected with an approach to a static posture of the arm characteristic of the grasping moment. However, if the movement of the artificial wrist were to be initiated at such target instants, this strategy would involve somewhat slow manipulation, since once close to the grasping point, one would have to wait for the wrist to position properly. Furthermore, it could only be used to orient the wrist in static postures. The advantage of this strategy would be the simplicity of the network and the limited amount of data to be recorded. The results obtained with the FFNN indicate that it is possible to achieve correlation coefficients of around 0.85 by training with data from several subjects and manipulating two different objects between varying locations. Correlations can be improved by introducing object information to up to about 0.95, although this would require computer vision systems for practical implementation. The results also indicate that, if the network is trained in a subject-specific manner, it is possible to improve the correlation.

The second strategy involved the use of a time-delay neural network (TDNN), which used data from a set of previous instants to predict the wrist turns at each instant. Unlike the FFNN, this network used input data recorded from a sequence of previous instants, so it had a higher predictive capability and also allowed dynamic prediction such that the prosthetic wrist could move at each instant depending on the past recorded input data. In this work, networks with time delay amplitudes of 0.1s, 0.5s, or 0.75s were tested, showing a significant improvement from 0.1s to 0.5s and little appreciable improvement when increasing from 0.5s to 0.75s. Correlation coefficients of up to 0.94 were achieved with the TDNN, taking into account that, in this case, the correlation was performed for the angular data from all the instants and not only for the PK and PL instants. Figure 11 shows the ability of these networks to predict dynamic wrist rotation as a function of elbow and shoulder angles in the tested transport tasks. The prediction presented similar levels of approximation for the cylindrical object and for the spherical object. The drawbacks of the TDNN compared to the FFNN in applications in prosthetic wrists are its higher complexity and the need to process more data, with consequent effects on the memory and computational capacity of the processors required for its implementation. However, as shown in Figure 12, reducing the delay time from 0.75 s to 0.1 s with a 20-neuron network would allow a considerable reduction in the number of network parameters at the cost of a not very significant loss in performance.

The results indicate that the mean angular errors in the predictions of both the FFNN and TDNN were approximately proportional to the range of motion of each degree of freedom, being approximately between 10% and 20% of the range of mobility. The errors for the FFNN trained with all data were less than 5° for F/E motion, 4° for A/A motion, and 9 for P/S motion. In the case of the TDNN, the error was close to 6° for F/E, 4° for A/A, and 12° for P/S.

As a limitation of this paper, it should be considered that the networks obtained would be valid only for transport tasks similar to those in the experiment. It is expected that, for other tasks, the relationships between arm movements and wrist movements would change, so a network trained for a particular task would probably not work correctly for other tasks. In this sense, the use of kinematic information from the arm is more limited than the use of other signals, such as EMG or ultrasound, since in these cases, the data are based on physiological information directly related to the activity of the muscles involved in the movement of the wrist. The use of networks similar to those trained in this work in a prosthesis would therefore be limited to the type of activity for which the network was trained. However, for a practical implementation, it would be feasible to have a series of selectable programs for common tasks that the user could choose depending on the task to be performed at any given time.

In this work, the prediction of wrist movements was approached using information for the elbow and shoulder angles as input data. In the TDNN, since data from several previous instants were included as input, the network also indirectly considered the angular velocity and acceleration in the movements. Although the data recorded with the sensors used also included position, this information was not used as training data. The reason was that angle information would be easier to obtain using low-cost, commercial inertial sensors in future implementations. On the other hand, linear kinematic variables, such as the relative position between sensors or the linear relative velocity between them, are directly related to joint rotations since they are open kinematic chains in which the segments can be considered rigid for practical purposes. One of the major problems with inertial sensors is that they are based on signal fusion and integration of several sensors (magnetometer, accelerometer, gyroscope), and they are prone to producing drift errors due to noise accumulation in the signals [15]. However, some recently proposed algorithms make it possible to considerably ameliorate the problem in the case of obtaining human joint angles from inertial sensors placed on both sides of the joint [16], enabling the reduction of the errors to values in the order of 3°–4° in activities of more than five minutes duration.

For future studies, we can consider the possibility of implementing the proposed networks in a prosthetic hand with a motorized wrist and analyzing the improvement achieved, comparing the task times and compensatory movements required. It would also be interesting to extend this methodology to other specific tasks or to manipulation in the specific environments of daily living activities. In this study, we used simple shallow neural networks; i.e., FFNN and TDNN. The use of deep networks, such as CNN or LSTM, could be a line of future research to extend this methodology to a wider range of activities.

## 5. Conclusions

The results of this research allow us to conclude that it is possible to use neural networks to predict with reasonable accuracy the F/E, A/A, and P/S movements of the wrist from the angles rotated at the elbow and shoulder for a task of transporting two objects between different positions on a vertical shelf.

With a simple FFNN with 20 neurons in a single hidden layer, the required wrist postures at the instant of picking up the object and that of releasing it in a different position could be predicted from the measured angles at the elbow and shoulder with some anticipation (0.75s), resulting in a correlation coefficient of 0.88. With a TDNN with 20 neurons in a single hidden layer, the wrist angles could be predicted dynamically for the entire object transport motion from the angles measured at the elbow and shoulder during the 0.75s prior to each instant, with a correlation between the predictions and the actual angles of 0.94. The prediction errors for the FFNN trained with all data were less than 5° for the F/E movement, 4° for the A/A movement, and 9° for the P/S movement. In the case of the TDNN, the error was close to 6° for F/E, 4° for A/A, and 12° for P/S. Among the different training functions studied in this work, the function with Bayesian regularization backpropagation (*trainbr* in Matlab) was the one that showed the best results. Regarding the number of neurons used, a tendency towards the stabilization of the prediction at around 15–20 neurons was observed. The correlations obtained by the networks improved when information for the manipulated object was added or the network was trained separately with the specific cases for each object (0.94 for FFNN, 0.96 for TDNN). Likewise, the correlation improved when training the network in a subject-specific way.

## Figures and Tables

**Figure 1 biomimetics-08-00219-f001:**
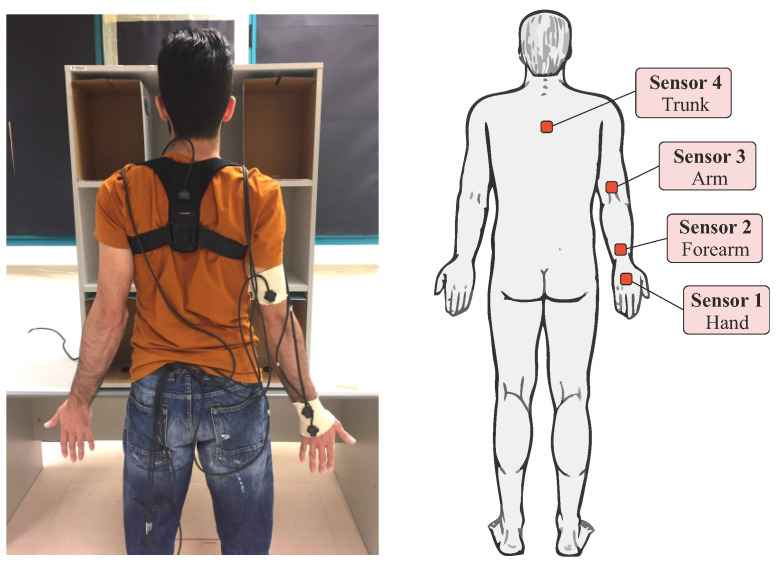
Placement of sensors on the subject.

**Figure 2 biomimetics-08-00219-f002:**
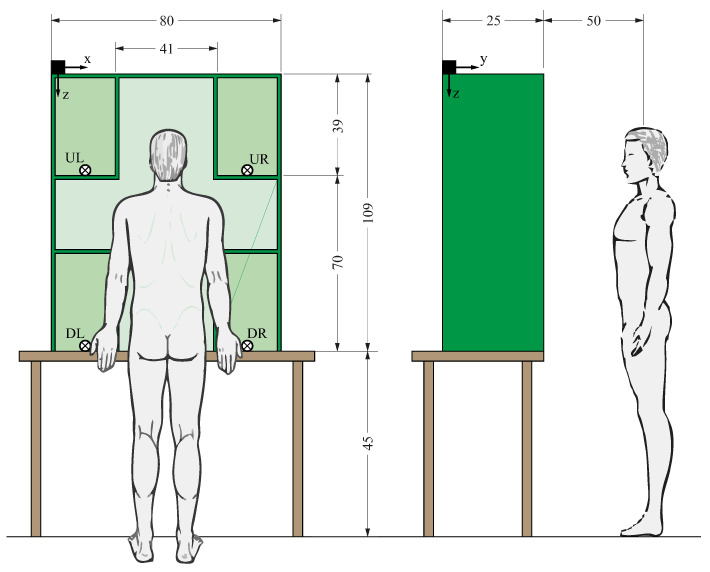
Arrangement of subject and objects during experiments (dimensions in cm).

**Figure 3 biomimetics-08-00219-f003:**
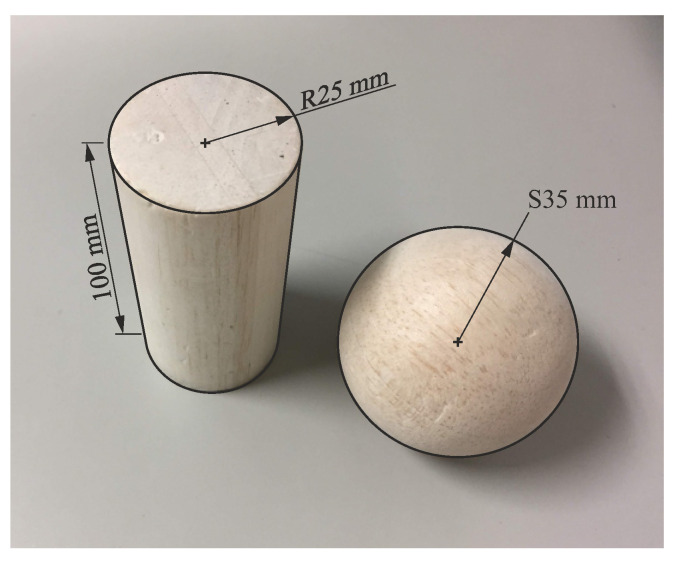
Objects of the SHAP protocol transported in the experiments.

**Figure 4 biomimetics-08-00219-f004:**
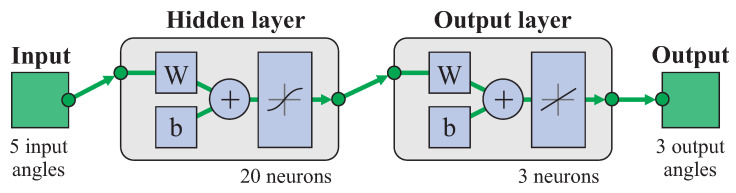
FFNN model used (example with 20 neurons in hidden layer, 5 input elbow and shoulder angles and 3 output wrist angles).

**Figure 5 biomimetics-08-00219-f005:**
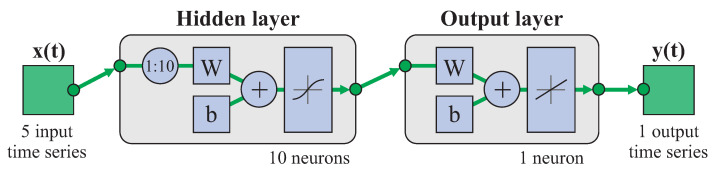
Model of TDNN used (example with 10 neurons in hidden layer, 1 time series for one output angle at the wrist, and 5 time series for input angles at the elbow and shoulder). In this case, the network used the previous 10 data points from the time series (1:10).

**Figure 6 biomimetics-08-00219-f006:**
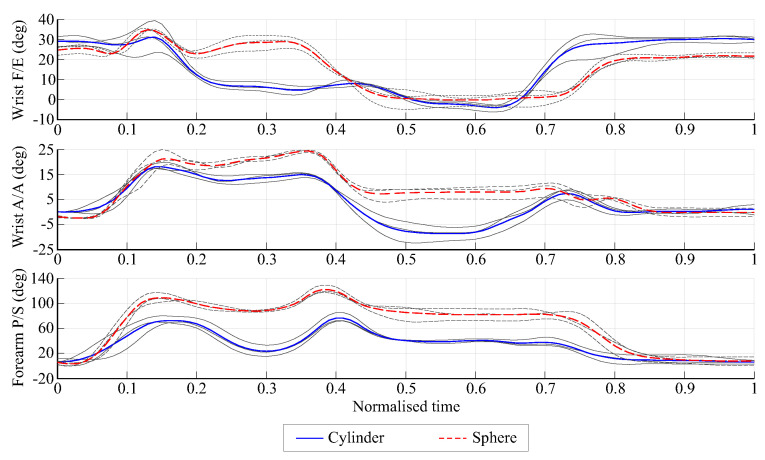
Angles of interest in the wrist for the movement of the cylinder and sphere from the UL to the DL position for subject one. The three repetitions (in gray) and the mean value (in color) are shown.

**Figure 7 biomimetics-08-00219-f007:**
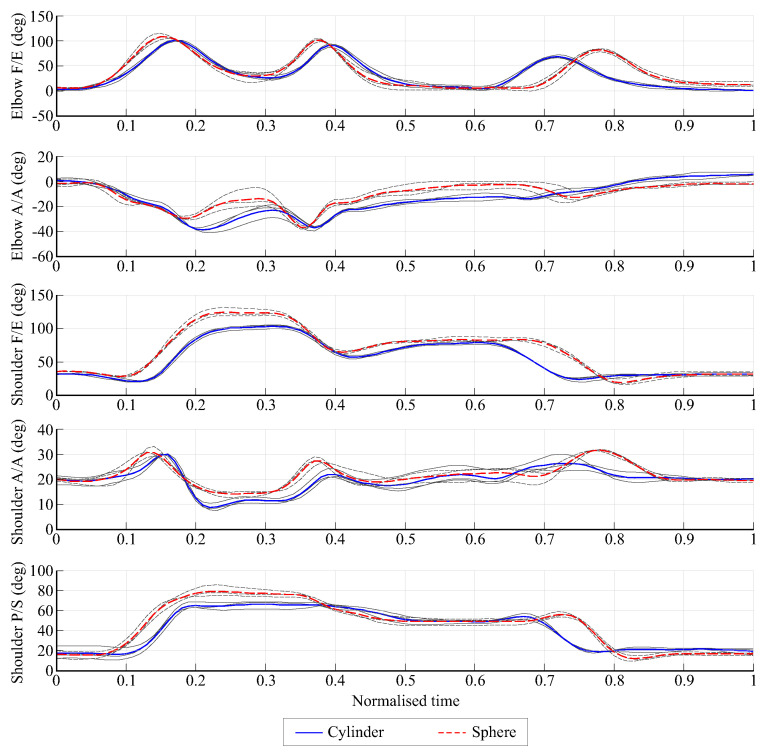
Angles in the elbow and shoulder for the movement of the cylinder and sphere from position UL to DL for subject one. The three repetitions (in gray) and the mean value (in color) are shown.

**Figure 8 biomimetics-08-00219-f008:**
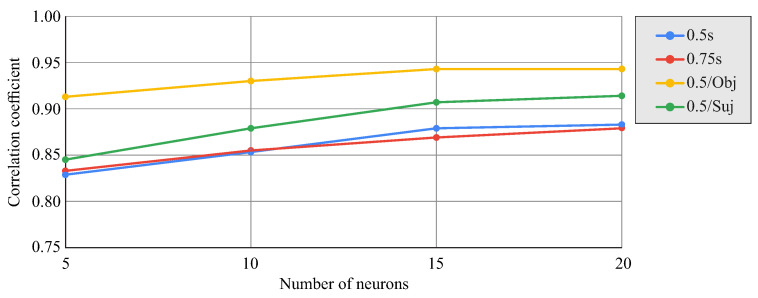
Correlation coefficients for the FFNN as a function of the number of neurons and the anticipation of the input data relative to the prediction instant (0.5s, 0.75s). The effect of adding the object or subject as input data for 0.5s anticipation is shown.

**Figure 9 biomimetics-08-00219-f009:**
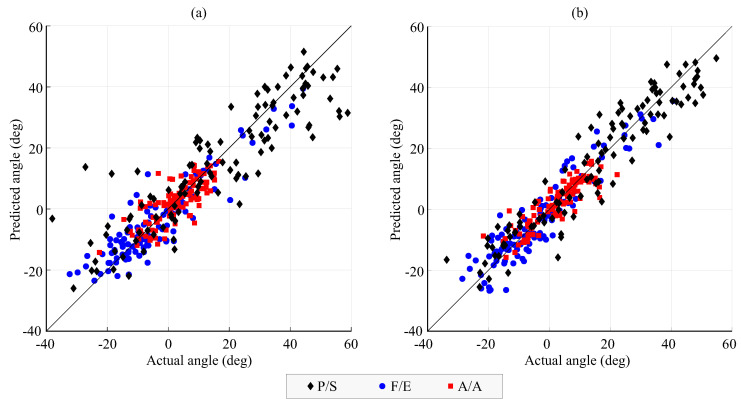
Angles predicted by the FFNN trained with the *trainbr* algorithm compared to the actual angle for 0.5s anticipation and 20 neurons without (**a**) and with (**b**) information for the transported object as input data. The angles represent variations from the mean joint angle for each subject.

**Figure 10 biomimetics-08-00219-f010:**
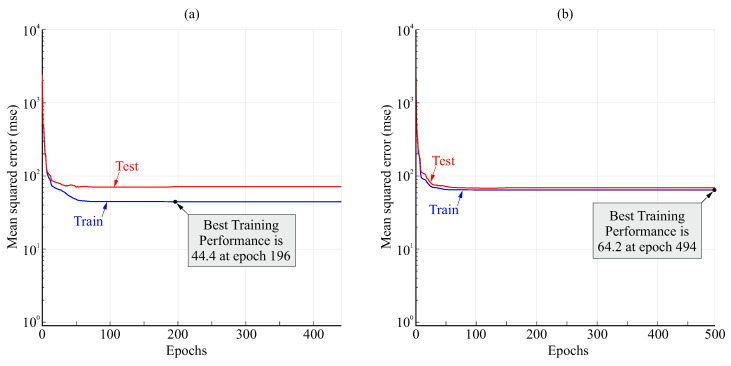
Examples of variation in the loss (mean squared error) for the FFNN for 0.5s anticipation and 20 (**a**) and 10 neurons (**b**).

**Figure 11 biomimetics-08-00219-f011:**
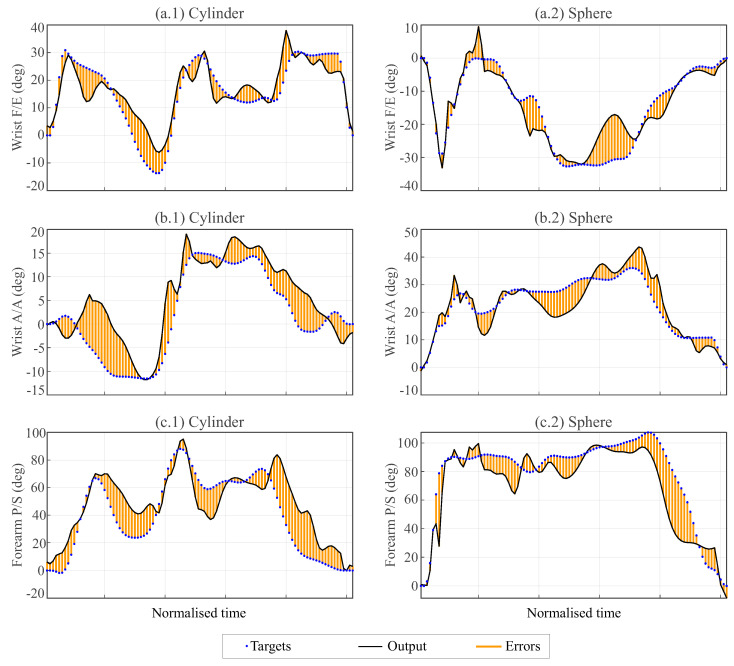
Wrist F/E (**a**) and A/A (**b**) and forearm P/S (**c**) angles estimated by the TDNN (outputs) and real values (targets) during the transportation of a cylinder (left) and sphere (right). The network included 15 neurons and a temporal anticipation amplitude of 15 instants (0.75s) in the input data. The training of the network was performed using the *trainbr* algorithm. The cases shown are a random selection of the test set.

**Figure 12 biomimetics-08-00219-f012:**
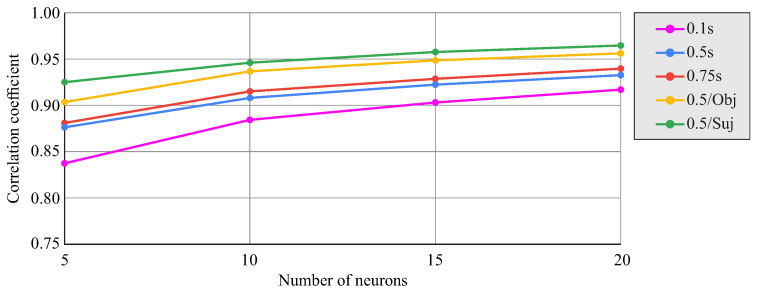
Correlation coefficient for the TDNN as a function of the number of neurons and the time delay amplitude (0.1s, 0.5s, 0.75s). The effect of restricting the data to only one object (average result for cylinder and sphere) or only one subject (subject one) with anticipation of 0.5s is shown.

**Figure 13 biomimetics-08-00219-f013:**
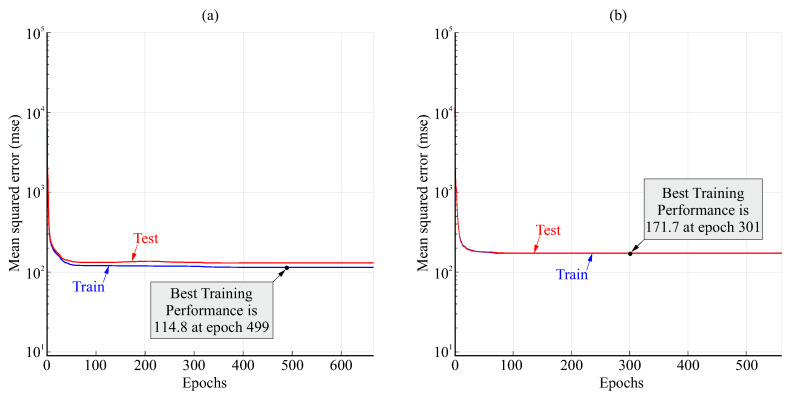
Examples of variation in the loss function (mean squared error) for the TDNN with anticipation of 0.5s and 20 (**a**) and 10 neurons (**b**).

**Table 1 biomimetics-08-00219-t001:** Mean error and correlation coefficient (CC) in the prediction of wrist angles as a function of the anticipation between data and prediction and the number of neurons in the hidden layer of the FFNN trained with the *trainbr* algorithm.

Anticipation (s)	# of Neurons	F/E Error (deg)	A/A Error (deg)	P/S Error (deg)	CC
0.50	5	5.74	4.25	11.23	0.829
0.50	10	5.73	4.23	10.2	0.853
0.50	15	5.29	3.91	8.75	0.879
0.50	20	4.88	3.80	8.85	0.883
0.75	5	6.47	4.63	10.30	0.833
0.75	10	5.83	4.33	9.67	0.855
0.75	15	5.38	3.76	9.52	0.869
0.75	20	4.91	3.55	8.98	0.879

**Table 2 biomimetics-08-00219-t002:** Error in the prediction of angles and correlation coefficients (CCs) between actual values and those estimated by the FFNN (anticipation 0.5 s, 20 neurons in hidden layer) with different training functions.

Function	F/E Error (deg)	A/A Error (deg)	P/S Error (deg)	CC
*trainbr*	4.88	3.80	8.85	0.883
*trainlm*	5.19	3.76	9.40	0.868
*trainscg*	6.27	4.38	12.60	0.803
*trasinbfg*	5.89	4.44	12.05	0.807

**Table 3 biomimetics-08-00219-t003:** Prediction error of the TDNN (trained with the *trainbr* algorithm) and the average correlation coefficient between the actual values and the predictions with different numbers of neurons and amplitudes of anticipation in the input data.

Anticipation (s)	# of Neurons	F/E Error (deg)	A/A Error (deg)	P/S Error (deg)	CC
0.50	5	8.99	5.89	15.54	0.876
0.50	10	7.59	5.38	13.03	0.908
0.50	15	6.51	4.53	12.46	0.922
0.50	20	6.20	4.09	12.81	0.938
0.75	5	8.77	5.21	15.50	0.881
0.75	10	7.41	4.88	14.28	0.915
0.75	15	7.06	4.57	16.30	0.929
0.75	20	6.30	4.13	12.51	0.940

**Table 4 biomimetics-08-00219-t004:** Error in the prediction of angles and mean correlation coefficients (CCs) for the TDNN (delay amplitude of 0.5 s and 20 neurons in hidden layer) with different training functions.

Function	F/E Error (deg)	A/A Error (deg)	P/S Error (deg)	CC
*trainbr*	6.20	4.09	12.81	0.938
*trainlm*	7.38	4.86	14.81	0.924
*trainscg*	7.89	4.83	15.95	0.891

**Table 5 biomimetics-08-00219-t005:** Mean absolute errors between actual angles (for F/E, A/A, and P/S) and those predicted by the linear regression models and the corresponding correlation coefficients.

Target Angle	Error (deg)	CC
F/E	11.70	0.35
A/A	7.75	0.43
P/S	21.97	0.74

## Data Availability

The data used for the study are available upon request from the authors.

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
