# Peer review of "Predicting Wrist Joint Angles from the Kinematics of the Arm: Application to the Control of Upper Limb Prostheses"

_biomimetics, 2023, doi:10.3390/biomimetics8020219_

Round 1

Reviewer 1 Report

The authors wanted to test their hypothesis that “for certain tasks to automate wrist movement based on the movement of other arm joints.” This is an interesting study. I believe this manuscript can be approved by addressing the following questions

·        The subjects had been trained or had been asked to limited variance, i.e., produce the same trajectories among repetitive tests. In Fig. 6, the variance in three repetitions was very small

·        From the data presented that has no strong nonlinear characteristics, the authors should provide linear regression results for comparison.

·        Were the NN models task-dependent? How about the performance when using the PICK-trained model, to test the PLACE task?

Author Response

We attach a point-by-point response in a document.

Reviewer 2 Report

This paper carries out the research of predicting wrist rotations in pick & place tasks based on kinematic information through neural networks. The work is interesting and meaningful, and it can be published after revision. Some suggestions are given as follows:

1. Why the author didn’t establish an analytical kinematical model?

2. Did the author consider the workspace constraint?

3. Why the author didn’t use deep neural networks? Such as CNN, LSTM.

4. The variation of loss value during train process should be given.

5. The references of the paper are not enough, in fact, the kinematic analysis of complex mechanisms is very important, such as the upper limb. The following papers about the kinematics of complex mechanisms (i.e. parallel manipulators) can be used to emphasize the importance of kinematics.

1) Minimum-time trajectory planning and control of a pick-and-place five-bar parallel robot, IEEE/ASME Transactions on Mechatronics, 2015, 20(2): 740–749.

2) A postprocessing strategy of a 3-DOF parallel tool head based on velocity control and coarse interpolation. IEEE Transactions on Industrial Electronics, 2018, 65(8): 6333-6342.

Author Response

(The authors gave the same response as above.)

Round 2

Reviewer 1 Report

The authors have addressed my previous questions

Reviewer 2 Report

no comments